



# The effect of strong shear on internal solitary-like waves

Marek Stastna[1], Aaron Coutino[1], and Ryan K. Walter[2]

[1]Department of Applied Mathematics, University of Waterloo, Waterloo, ON, Canada, N2L 3G1.
[2]Physics Department, California Polytechnic State University, San Luis Obispo, CA, USA

**Correspondence:** Marek Stastna (mmstastn@uwaterloo.ca)

**Abstract.** Large amplitude internal waves in the ocean propagate in a dynamic, highly variable environment with changes in background current, local depth, and stratification. The Dubreil-Jacotin-Long, or DJL, theory of exact internal solitary waves can account for a background shear, doing so at a cost of algebraic complexity and a lack of a mathematical proof of algorithm convergence. Waves in the presence of shear that is strong enough to preclude theoretical calculations have been reported in observations. We report on high resolution simulations of stratified adjustment in the presence of strong shear currents. We find instances of large amplitude solitary-like waves with recirculating cores in parameter regimes for which DJL theory fails, and of wave types that are completely different in shape from classical internal solitary waves. Both are spontaneously generated from general initial conditions. Some of the waves observed are associated with critical layers, but others exhibit a propagation speed that is very near the background current maximum. As such they are not freely propagating solitary waves, and a DJL theory would not apply. We thus provide a partial reconciliation between observations and theory.

## 1 Introduction

Large amplitude internal waves, often referred to as internal solitary–like waves or ISWs, are a well–studied coherent, nonlinear phenomenon accessible via field measurements, laboratory experiments, and simulations of density stratified fluids. Historically, ISWs were described by approximate, perturbation theories that lead to an equation of Korteweg-de Vries (KdV) type for the temporal-horizontal portion of the object (Helfrich and Melville (2006); Talipova et al. (1999)). The vertical component is described by solving a linear ordinary differential eigenvalue problem. This is generally referred to as the weakly nonlinear description, or WNL. While aspects of WNL continue to be actively developed in recent literature (e.g. Talipova et al. (1999); Grimshaw et al. (1997); Caillol and Grimshaw (2012)), it has been known for some time that WNL is lacking as a quantitative descriptor of ISWs. The KdV theory itself predicts that wave amplitude is bounded above by the onset of breaking, in contrast to observations of wave broadening and the formation of waves with flat crests (i.e., tabletop waves; Lamb and Wan (1998); Rusås and Grue (2002)). Exact ISWs are solutions of the nonlinear elliptic eigenvalue problem referred to as the Dubreil-Jacotin-Long (DJL) equation. This equation is formally equivalent to the full stratified Euler equations (in a frame moving with the wave), and thus this theory is referred to as fully nonlinear, or exact.

Comparisons of WNL with exact ISWs (Lamb and Yan (1996); Lamb (1998); Stastna and Peltier (2005)) typically yield poor results for the vertical structure, especially for complex stratifications. Higher order WNL theory, with the Gardner or mKdV equations as prominent examples, does a better job of qualitatively matching the type of upper bound on wave amplitude





(Grimshaw et al. (1997)), but there is a significant gap between what is observed in simulations and what WNL can describe (Stastna and Peltier (2005)). In the context of ISWs with strong shear, this gap is well illustrated by the study of (Caillol and Grimshaw (2012)), which develops a WNL theory for weakly stratified critical layers. The theory is algebraically quite

complex with significant ties to work on Rossby wave critical layers. However, the theory is not compared to simulations or experiments, and thus it is unclear to what extent it applies to a situation with a dynamically evolving wave field. Interestingly it is very similar in style to Maslowe (1980) which considered a closely linked topic, some 30 years earlier.

While field measurements are often taken in a dynamic, complex environment with changes in the stratification structure and a non-trivial background current, much of the understanding of ISWs has been developed based on relatively simple

stratifications (typically quasi-two layer, or exponential stratification). The presence of a background shear current complicates the algebra needed to derive theory, whether WNL or the DJL equation. In the case of WNL, the numerical methods are unchanged, but for the DJL equation the iterative algorithm used to obtain solutions requires an ad hoc modification (see Stastna and Lamb (2002) and the Methods section below). In practice, publicly available software for the DJL equation (Dunphy et al. (2011)) can handle many situations, but there is no *a priori* proof of convergence. It has been shown that the presence of a

background shear current can modify the type of upper bound on wave amplitude, and can also change the polarity of exact ISWs (again see Stastna and Lamb (2002) for details). The detailed oriented reader is cautioned that Figure 3 in Stastna and Lamb (2002) is based on ISWs of elevation which have an opposite sign of wave–induced voritcity to ISWs of depression. For ISWs of depression a background current with positive vorticity tends to decrease the limiting wave amplitude. In his study of ISW energetics in the presence of a background shear current Lamb (2010) thus employed a negative background current.

The DJL equation has also been extended to model ISWs past breaking, or waves with trapped cores. Such waves are known to form during shoaling (Lamb (2002)), and the question of whether the core is completely trapped (Xu et al. (2016)), quiescent: (Derzho and Grimshaw (1997); Luzzatto-Fegiz and Helfrich (2014); Lamb and Wilkie (2004)), or found immediately adjacent to the boundary or at depth (He et al. (2019)) have all been the subject of recent studies. Any ISW can be described as a propagating baroclinic vortex, but trapped cores provide a more complex wave–vortex coupling that will be revisited below.

In the classical theory of hydrodynamic stability the gradient Richardson number ($Ri$) is the standard necessary condition; when $Ri > 0.25$ the flow is linearly stable. However, $Ri < 0.25$ is not a sufficient condition for instability, (Hazel and Hazel (1972)). Large amplitude ISWs, both with and without a background shear current, can induce strong shear currents near the wave crest. This situation has drawn interest over several decades (Bogucki and Garrett (1993); Barad and Fringer (2010); Lamb and Farmer (2011); Xu et al. (2019)) first as a pure conjecture, then with two-dimensional and finally three-dimensional

situations. It has been found that shear instability within ISWs is quite complex, with onset dependent on the strength of upstream perturbations, the Reynolds number, and duration of time spent below $Ri = 0.25$ as a perturbation passes through the wave. Three-dimensionalization, at least on scales for which direct numerical simulation (DNS) has been accessible, occurs preferentially at the rear of the wave and in some parameter regimes instability is episodic with bursts and quiescent periods.

Walter et al. (2016) measured large amplitude ISWs in northern Monterey Bay, California (USA) as part of a complex

interplay between a coastal upwelling front and and local wind forcing. These waves were large amplitude (up to a 10 m maximum isopycnal displacement in water approximately 20 m deep) with strong background shear currents. Indeed the





background shear currents were so strong that solutions of the DJL equation were impossible to compute. Some analysis of large ISW–like waves in the presence of strong background shear currents is provided in Stastna and Walter (2014). These authors considered the resonant generation of ISWs by flow of a background current with shear over isolated topography. They

found large amplitude wave trains, as well as waves trapped behind the topography with vortex rich cores that were quasi-trapped near the surface (i.e., both the upstream propagating wavetrains and the trapped waves exhibited vortex-rich cores). However, the connection between simulations and field measurements is incomplete since it is unclear if the waves observed in Walter et al. (2016) were resonantly generated.

In recent work, Zhang et al. (2018) considered the effect of a background current on mode-2 waves. The waves were

generated via stratified adjustment, similar to the methodology we follow below. The stratification used was centered at the mid-depth, so that in the absence of a shear current initial perturbations have the dominant part of their energy deposited into the mode-2 wave field. The authors document the manner in which the adjustment process, as well as the resulting leading mode-2 wave and trailing mode-1 tail are modified by the presence of shear. They also compare their results to the observations of Shroyer et al. (2010), though the fact that both the density profile and shear layer in Zhang et al. (2018) are centered at the

mid-depth makes a detailed comparison impossible.

The interaction of ISWs with background currents has also been explicitly demonstrated in the context of frontal generation. Bourgault et al. (2016) discussed observations and simulations of a case in the Saguenay fjord, Quebec, Canada. The authors found that convergence near the front was important in numerical generation experiments, though the parameter space they explored was fairly limited. In all cases discussed by the the authors, rightward and leftward propagating wavetrains propagating

away from the front differ in amplitude, with larger waves observed in stronger shear.

In a related non–ISW context, Lamb and Dunphy (2018) considered the generation of internal waves by tidal flow in the presence of shear. They concentrated on characterizing energy flux, and found profound asymmetries between the waves propagating with the shear and against the shear. Interestingly, whether the downstream or upstream side of the ridge that generated the internal waves dominated the energy flux was found to depend on the slope of the flanks of the ridge. It is

difficult to ascertain implications of this study for the problem of ISWs with shear, since these authors only considered a linear stratification. Nevertheless, the study clearly suggests that shear can profoundly influence wave energetics.

In this manuscript, we report on high resolution simulations of stratified adjustment and the resulting internal wave field in the presence of strong shear currents. The structure of the manuscript is organized as follows. The Methods section presents the governing equations, the DJL theory and the design of, and parameter values for, all numerical experiments we present.

The Results section presents the major findings, divided into three parts: i) waves modified by shear but without critical layers, ii) waves with critical layers, iii) situations in which the stratification is in the opposite half of the water column from the shear layer. The manuscript concludes with a Discussion section and a brief set of Conclusions.



## 2  Methods

We consider an incompressible fluid in the absence of rotation that obeys the Boussinesq and rigid lid approximations. The
stratified Navier Stokes equations in the absence of rotation read

$$\frac{\partial \boldsymbol{u}}{\partial t} + \boldsymbol{u} \cdot \boldsymbol{\nabla} \boldsymbol{u} \;=\; -\frac{1}{\rho_0}\boldsymbol{\nabla} P + \nu \nabla^2 \boldsymbol{u} - \frac{\rho g}{\rho_0}\hat{k}, \tag{1}$$

$$\boldsymbol{\nabla} \cdot \boldsymbol{u} \;=\; 0, \tag{2}$$

$$\frac{\partial \rho}{\partial t} + \boldsymbol{u} \cdot \boldsymbol{\nabla} \rho \;=\; \kappa \nabla^2 \rho, \tag{3}$$

where $\boldsymbol{u}$ is the velocity, $P$ is the pressure, $\rho$ is the density, and $\rho_0$ is some reference density of the fluid. The physical parameters
are the shear viscosity $\nu$ (set to equal $1 \times 10^{-6}$ m s$^{-2}$) and scalar diffusivity $\kappa$ (set to equal $2 \times 10^{-7}$ m s$^{-2}$). The unit vector
in the vertical direction is denoted by $\hat{k}$. The computational domain is rectangular with the x-axis running left to right along
the bottom of the domain, so that $0 < x < L_x$ and $0 < z < L_z$.

Simulations were carried out with the pseudospectral code SPINS (Subich et al. (2013)). The code has been thoroughly
validated in a number of different configurations (e.g. shear instabilities, internal wave generation, internal solitary wave prop-
agation) and is available for download through its online manual:

```
https://wiki.math.uwaterloo.ca/fluidswiki/index.php?title=SPINS_User_Guide
```

All simulations reported employed regularly spaced grids. Approximately 30 exploratory simulations were carried out, prior
to identifying a parameter space of interest. Table 1 provides a list of cases discussed in this manuscript, along with their key
physical and numerical parameters. As a general comment, simulations were much higher resolution than many comparable
studies in the literature, and the pseudospectral aspect of SPINS means that the model has very little numerical dissipation.
This proved vital for some of the results reported below, and is discussed further in the Discussion section.

All simulations were initialized with a background shear current

$$U(z) = U_{max} \exp[(z - L_z)/d_U]. \tag{4}$$

The total depth was fixed as $L_z = 20$ m for all simulations, but the horizontal extent of simulations was varied on a case by
case basis. The thickness of the shear layer was fixed at $d_U = 3$ m, or $0.15$ of the total depth. $U_{max}$ was varied as described
in Table 1. The form of the background current is such that, in the absence of stratification, it is linearly stable (i.e. Fjortoft's
criterion is not satisfied).

The initial density field was specified as $\rho(x,z) = \bar{\rho}(z - \eta)$ with

$$\eta = \eta_0 \exp[-(x - 0.5Lx)^2/w_d^2]. \tag{5}$$

While other values were used in preliminary experiments, all numerical experiments reported below set $\eta_0 = -5$ m ($0.25$ of
the total depth) and $w_d = 100$ m.

The background density field was specified as

$$\bar{\rho}(z) = 1 - \frac{\Delta \rho}{2} \tanh[(z - z_0)/d]. \tag{6}$$





The dimensionless density difference was set as $\Delta\rho = 0.001$ for all simulations, while the pycnocline center and thickness,
$(z_0, d)$, were varied as shown in Table 1.

Figure 1 shows the one dimensional background profiles for the various simulations discussed below. Panel (a) shows the $N^2(z)$ profile, panel (b) shows the background shear current $U(z)$, and panel (c) shows the gradient Richardson number ($Ri$).

Exact internal solitary waves are solutions of the Dubreil Jacotin Long (DJL) equation. In the presence of a background shear current this equation has a rather complex form

$$\nabla^2 \eta + \frac{U_z(z-\eta)}{[U(z-\eta)-c]}\left[1 - \left(\eta_x^2 + (1-\eta_z)^2\right)\right] + \frac{N^2(z-\eta)}{[U(z-\eta)-c]^2}\eta = 0. \tag{7}$$

For no background current the DJL equation takes the more familiar form

$$\nabla^2 \eta + \frac{N^2(z-\eta)}{c^2}\eta = 0$$

with the strong non–linearity encapsulated in the evaluation of the buoyancy frequency squared profile at its upstream height (i.e. at $z-\eta$). While several different techniques for the solution of the DJL equation without a background current are described
in the literature, the only publicly available approach to the DJL equation with a background current that the present authors are aware of employs an *ad hoc* extension of the direct variational method of Turkington et al. (1991), as described in Stastna and Lamb (2002). This consists of an iterative algorithm, which while generally stable for weak currents, leads to 'wandering' when shear is strong. The simulations reported below specifically concentrate on parameter regimes in which the DJL was found not to converge (unless otherwise noted in the discussion).

| Label | Description | $U_{max}$ (m s$^{-1}$) | $(z_0, d)$ (m) | Domain Size (m) | Grid Size |
|---|---|---|---|---|---|
| DJL | Shear with DJL Theory | 0.15 | (15.0,3.0) | (4000,20) | (8192,512) |
| DJLB | Shear with DJL Theory 2 | 0.15 | (5.0,3.0) | (4000,20) | (8192,512) |
| NO DJL 1 | Shear without DJL Theory 1 | 0.3 | (15,3.0) | (16000,20) | (16384,512) |
| NO DJL 1B | Shear without DJL Theory 1b | 0.3 | (5.0,3.0) | (4000,20) | (8192,512) |
| NO DJL 1BF | Shear without DJL Theory 1b | 0.3 | (5.0,3.0) | (4000,20) | (8192,512) |
| NO DJL 2 | Shear without DJL Theory 2 | 0.4 | (15,0.6) | (16000,20) | (16384,512) |
| CL 1 | Shear without DJL Theory 3 | 0.5 | (15.0,3.0) | (16000,20) | (16384,512) |
| CL 1B | Shear without DJL Theory 3 | 0.5 | (15.0,3.0) | (4000,20) | (8192,512) |
| CL 2 | Shear without DJL Theory 4 | 0.5 | (15,0.6) | (4000,20) | (32768,512) |

**Table 1.** Cases considered throughout the manuscript, including case label, and their dimensional parameters.

## 2.1 Results

Simulations are reported for combinations of background profiles shown in Figure 1. In all cases, the region of highest shear occurs away from the peak in the buoyancy frequency. For cases NO DJL 2 and CL 2 this is due to the thin pycnocline, and for cases with the sub–label $B$ this is due to the fact that the pycnocline center is below the mid-depth.


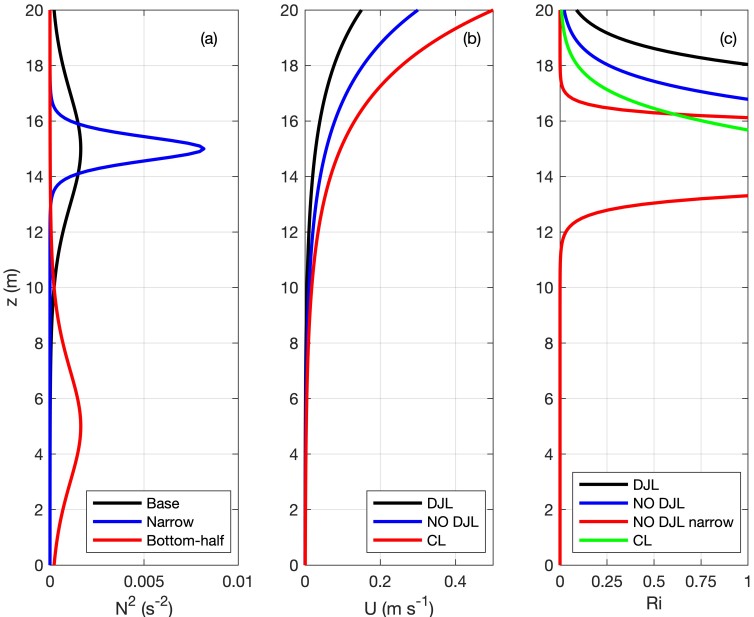

**Figure 1.** Background profiles. (a) $N^2(z)$, (b) $U(z)$, (c) $Ri$.

Numerical experiments were designed to compare three qualitative categories of results:

1. Those with wavetrains that have a DJL description propagating both with (rightward) and against (leftward) the background shear current; Cases DJL and DJLB.

2. Those with wavetrains that have a DJL description propagating against the background shear current (leftward) but no DJL description for wavetrains propagating with the background shear current (rightward); Cases NO DJL 1, NO DJL 1B, NO DJL 1BF, and NO DJL 2.

3. Cases with possible critical layers; Cases CL 1, CL 1B, CL 2.

For Cases DJL and DJLB, the background shear current was not large enough to preclude DJL solutions for either the rightward or leftward propagating wavetrains. Figure 2 shows the leftward and rightward propagating wave trains at $t = 7,200$ s for cases DJL and DJL B.

In both cases a rank ordered wavetrain is formed, with the leading wave described by solutions of the DJL equation. When
the pycnocline is in the upper half of the domain, waves of depression form, and the asymmetry between steeper and taller leftward propagating (against background shear) waves and broader, shorter rightward propagating (with background shear) waves is clearly evident. When the pycnocline is in the lower half of the domain, waves of elevation formed, the left–right asymmetry is less pronounced, and the fissioning of a rank ordered wave train takes place more slowly.



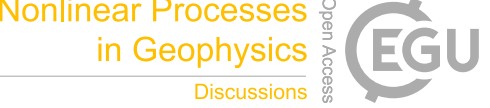
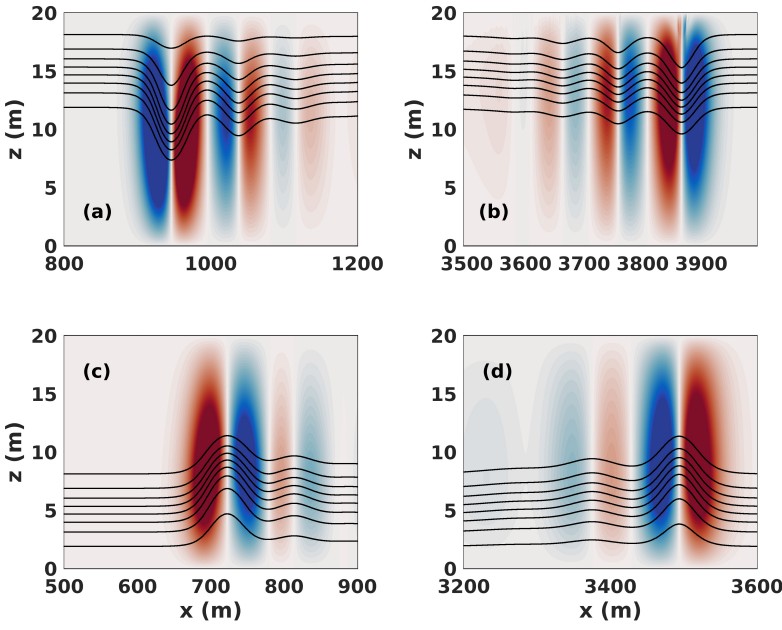

**Figure 2.** Vertical component of the velocity shaded and saturated at $\pm 0.01$ m s$^{-1}$ and eight density contours in black at $t = 7,200$ s for Cases DJL and DJLB. (a) Case DJL leftward (against shear) propagating wavetrain and (b) Case DJL rightward (with shear) propagating wavetrain. (c) Case DJLB leftward (against shear) propagating wavetrain and (d) Case DJLB rightward (with shear) propagating wavetrain.

## 2.2 Solitary-like wavetrains without DJL theory

We now turn to the more dynamically interesting cases for which DJL theory proves incomplete. We first consider the case labelled NO DJL 1. The stratification corresponds to the black curve in Figure 1a, while the background shear current corresponds to the blue curve in Figure 1b. This case yields coherent wavetrains propagating both with and against the background shear current, but the iterative DJL solver is not able to converge to steady solutions for the rightward propagating waves going with the background shear current. This is almost certainly due to the low values of $Ri$ in the shear layer, as indicated by the

blue curve in Figure 1c.

Figure 3 shows the density field in panel (a), and the vertical component of velocity, or $w$, in panel (b). The response of the density field at this advanced time ($t = 21,600$ s) is clear: two wavetrains form, with propagation distance altered by the Doppler shift due to the vertical mean of the background current. A large amplitude wave train propagates against the shear current (waves are leftward propagating). Its constituent waves are rank-ordered with no sign of instability. The portion of the

domain between $8 < x < 12$ km is dominated by the remnant of the adjustment process, or the portion of the initial disturbance that is deformed by the background shear current to form an overturning region. The wavetrain propagating with the shear





current (waves are rightward propagating) can be seen for $x > 14$ km. Again the waves appear rank-ordered, but are smaller in amplitude and much wider than the waves that make up the leftward propagating wavetrain moving against the shear current.

Figure 4 revisits the above comment on stability, but at an earlier point in the evolution of the wavetrains, namely at $t = 7200$
s. Panel (a) shows the $N^2(z)$ field using a symmetric color range, whereby the blue regions denote static instability ($N^2 < 0$). It can be seen that remnants of the initial adjustment yield a very large region of overturns, but interestingly some overturning extends to very near the front of the rightward (or with the background shear current) propagating wavetrain. Panel (b) shows $Ri(z)$, saturated over a color range $-0.25 < Ri < 0.25$. It can be seen that the leftward propagating (or against the background shear current) wave train only yields regions with $Ri < 0.25$ very near to the upper boundary. In contrast, the adjustment
remnants and rightward propagating wavetrains both yield regions of $Ri < 0.25$ over a region spanning several kilometers near the upper boundary.

The detailed evolution of the wave-trains, as expressed in the density field, is shown in Figures 5 and 6. Three times are shown ($t = 7, 200, 14, 400$ and $21, 600$ s), with a domain that extends 1 km in the horizontal, but shifts locations as the wavetrain propagates. If the reader was not told of its presence in Figure 5, they would have no reason to suspect that a background shear
current was present as the leftward propagating (against the shear) waves take the form of classical "bell-shaped" solitary waves of both WNL and DJL theories. This is in sharp contrast to rightward propagating (with the shear) waves in Figure 6. Here the region near the upper boundary is clearly active, with short wave and "roll" activity evident over a significant portion of the domain. The leading waves take a very different form from classical solitary waves, with sharp crests more akin to Stokes waves. Figure 7 reconsiders the near upper boundary region (top 2 m of the domain) for the wavetrain propagating
rightward with the shear current. Forty density isolines that range over $\rho_{min} < \rho < \rho_{min} + 0.2\Delta\rho$ are shown. By concentrating the view on this region, it can be seen just how active the near boundary region is. Not only do the rightward propagating waves exhibit quasi-trapped, recirculating cores that remain active over a long time period, it can be seen that a tumultuous region of overturns and rolls initially nearly reaches the two leading waves, but gradually lags behind. This figure clearly illustrates that the most important feature of internal solitary-like waves is their ability to outrun localized regions of overturns and instability,
before such regions drain significant energy from the wave. In the language of stability theory, nearly all instabilities that are spontaneously generated are convective (as opposed to global) in nature.

In Table 2, we show estimates of the propagation speed of the leading rightward propagating wave going with the background current. It is immediately evident that all cases discussed above have propagation speeds ($c_{est}$) that are very near the background current maximum ($U_{max}$). Moreover the scaled propagation speed ($c_{est}/U_{max}$) tends toward 1 as $U_{max}$ increases. The DJL
theory assumes that streamlines (in a frame moving with the wave) connect to infinity both upstream and downstream of the wave. This is not the case for the NO DJL cases. We hypothesize that this is why the DJL solvers fail to converge in this parameter range. Essentially, the wave trains that forms cannot be fully decoupled from the adjustment region for a long enough time so that a vortex dominated leading wave (or several waves) form.

To summarize, in all 'NO DJL' cases shown in Table 1, rightward propagating (with the background shear current) waves
with trapped cores and near-surface regions of overturns and local instabilities were consistently observed. These waves cannot


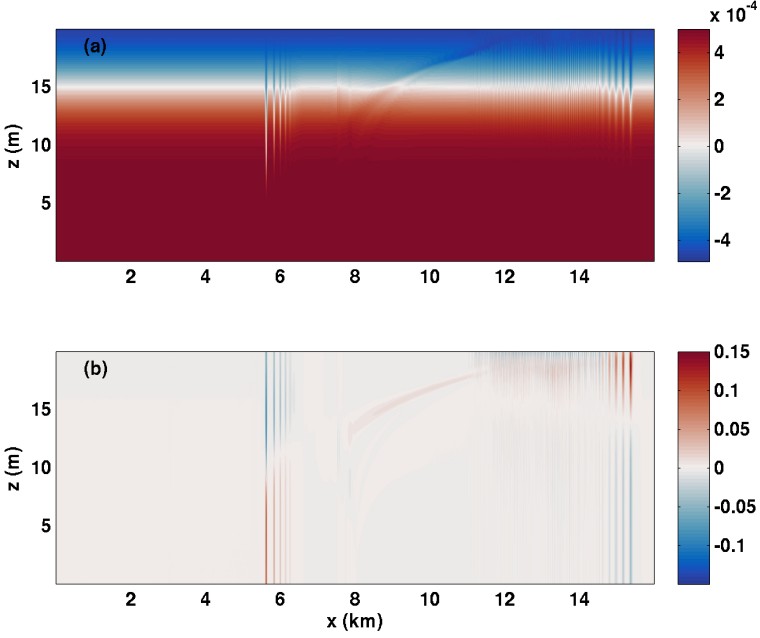

**Figure 3.** Sample wave trains at late time ($t = 21,600$ s) for the case NO DJL 1. (a) scaled density and (b) vertical component of velocity.

be described by presently available solvers for the DJL equation. In all simulations they were observed to be long lived, with active core regions that could transport mass over a considerable distance in the field.

### 2.3  Wave trains with critical layers

The waves described in the previous section (all 'NO DJL' cases) were found to have propagation speeds that were greater, if
only marginally, than the maximum background shear current. When $U_{max} > c > 0$ a critical layer may form. In order to get some sense of how robust the waves described above are to the presence of a critical layer we present the results of two cases with $U_{max} = 0.5$ m s$^{-1}$. In Figure 8 we show the vertical velocity and density fields over the top four meters for the wave train propagating rightward with the background current at three different times. The leading and second wave are shown, with both exhibiting a prominent quasi-trapped, recirculating core. Interestingly, while the quasi-trapped core of the leading wave
appears relatively quiescent, a region of instability is observed ahead of it near the top boundary.

The majority of the rightward propagating wavetrains described above involved trapped cores. In contrast, the cases with DJL theory yield non-breaking waves. In order to explore the effect of stratification in the high shear region we performed a series of experiments with a narrower pycnocline, and report one representative example, case CL 2. We found that the leading waves no longer matched the bell-shaped internal solitary waves of DJL theory, and that the short length scale behavior in the
high shear region required a substantial increase in horizontal resolution ($\Delta x = 0.12$ m for this case). In Figure 9 panel (a)


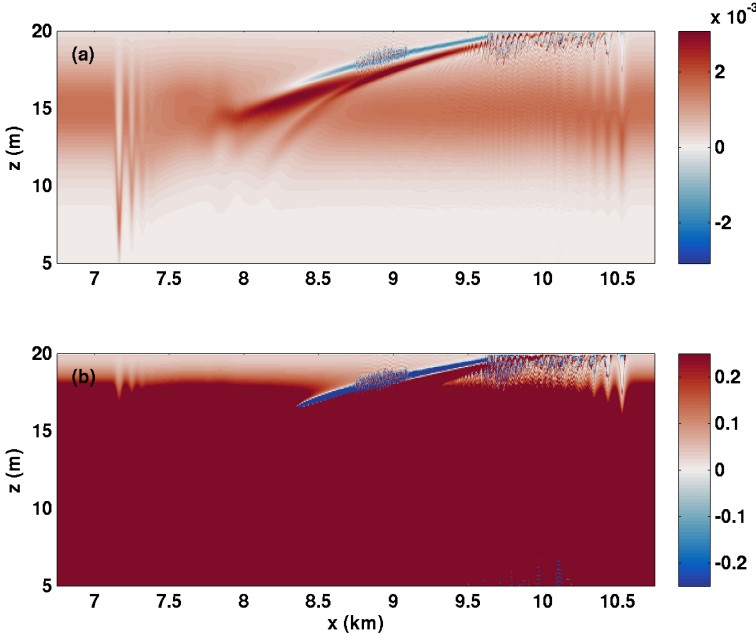

**Figure 4.** Case NO DJL 1. (a) sample $N^2$ and (b) $Ri$ at $t = 7,200$ s. For (b), color range saturated over $-0.25 < Ri < 0.25$.

we show the density field for the rightward propagating wave train. Note that the horizontal length is reported in meters, as opposed to kilometers for the previous figures. The leading wave can be seen to be horizontally asymmetric, with isopycnals in the region downstream of the crest that do not return to their upstream height. Panels (b) and (c) show the horizontal component of velocity saturated between $\pm 0.005$ m s$^{-1}$. Panel (b) corresponds to the region between thick black lines in panel

(a), while panel (c) corresponds to the region between thin dashed black lines in panel (a). Panels (b) and (c) show only the upper two meters of the water column. Panel (c) exhibits an inner region of strong vertical currents and a broader region of weaker currents. The outer region extends much further upstream of the wave crest (faint blue) than downstream (faint red). The leading wave is trailed by a long wavetrain. Panel (b) shows that the vertical currents in the wave train exhibit similar, short length scale fluctuations near the upper boundary (where the presumed critical layer is located).

The relationship between the rightward propagating wave train and the background current is better illustrated via the relative vorticity field, $\omega_{relative} = u_z - w_x - U'(z)$, where $U(z)$ is the background shear current. The relative vorticity field is shown in Figure 10. It can be seen that the leading wave has a substantial vortex that is trapped near the upper boundary. The wave train behind the leading wave is associated with a train of vortices that is gradually descending form the upper boundary into the main water column. Thus what is referred to as 'the leading wave' above is, to a substantial degree, a deformation of the

pycnocline by the leading vortex.


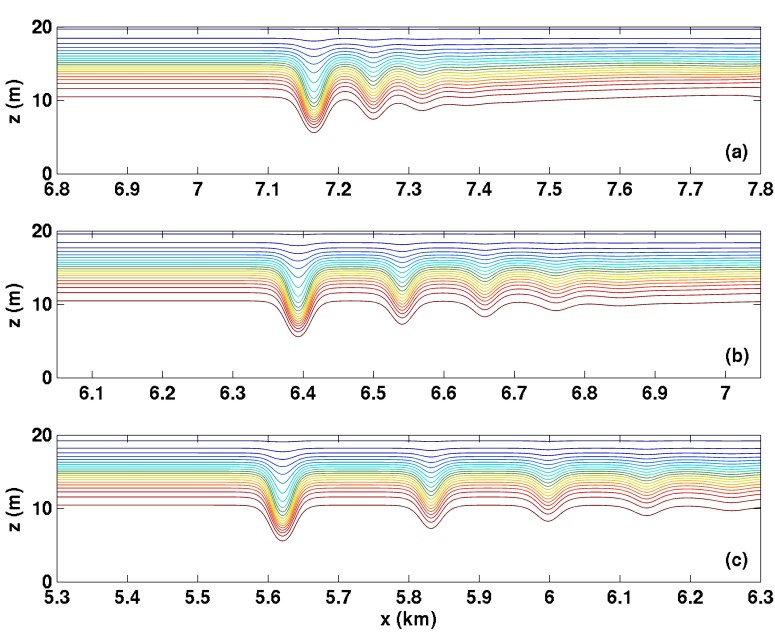

**Figure 5.** Density fields for wavetrains propagating against the shear current for the case NO DJL 1. (a) $t = 7,200$ s, (b) $t = 14,400$ s, and (c) $t = 21,600$ s.

| Label | estimated leading wave speed, $c_{est}$ (m s$^{-1}$) | $U_{max}$ (m s$^{-1}$) | $c_{est}/U_{max}$ |
|---|---|---|---|
| DJL | 0.252 | 0.15 | 1.68 |
| NO DJL 1 | 0.343 | 0.3 | 1.14 |
| NO DJL 2 | 0.408 | 0.4 | 1.02 |
| CL 1 | 0.48 | 0.5 | 0.96 |
| CL 2 | 0.483 | 0.5 | 0.97 |

**Table 2.** Estimated wave speeds for select cases.

### 2.4 Wavetrains with unstratified shear layers

The results in Figure 10 suggest that stratification in the shear layer region has a profound influence on the type of wave that forms via adjustment. In the absence of a shear layer, a pycnocline located in the bottom half of the water column yields ISWs of elevation. We thus performed a number of numerical experiments with a pycnocline located in the botom half of the water column to see if wave trains of elevation behave similarly to those reported above.

The results of our numerical experiments indicate that waves propagating against the background current are far less affected when the pycnocline is well removed from the shear layer. In contrast, ISWs propagating with the shear are almost completely

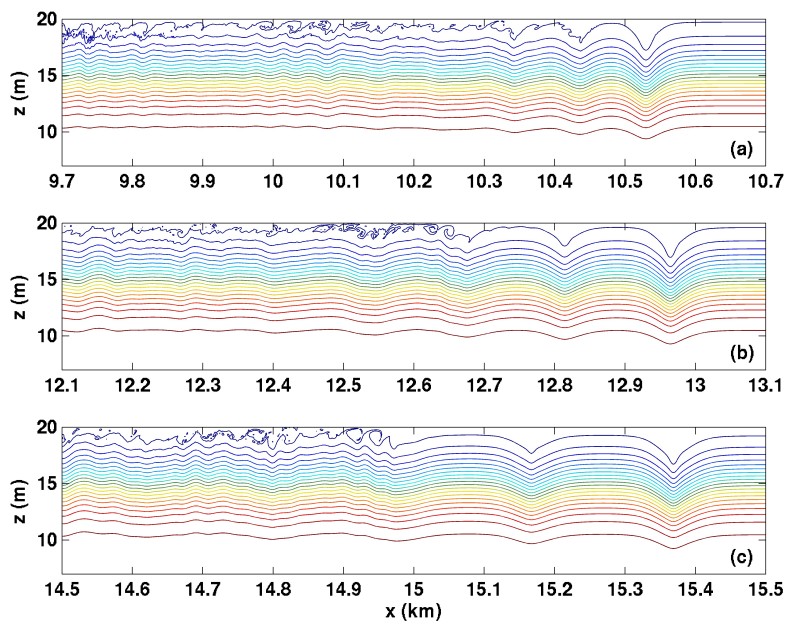

**Figure 6.** Density fields for wavetrains propagating with the shear current for the case NO DJL 1. (a) $t = 7,200$ s, (b) $t = 14,400$ s, and (c) $t = 21,600$ s.

destroyed in all but the weakest currents tried. Figure 11 shows the vertical component of the velocity field shaded with a saturation with 8 density isolines superimposed. The strength of the background shear current increases from top to bottom.

In the weakest shear current case [panel (a)] the trademark updraft-downdraft pattern of a DJL ISW can be seen for both the rightward and leftward propagating wavetrains. At the time shown, the wave train has not fully fissioned into rank–ordered ISWs. The reason for selecting such an early time in the evolution, $t = 5400$ s, can be seen in the bottom two panels. Panels (b) and (c) show $U_0 = 0.3$ and $U_0 = 0.5$ m s$^{-1}$, respectively. In both cases there is no trace of a coherent rightward propagating wave train, and the perturbation vertical velocity field is dominated by relatively short length scale perturbations. In panel (b)

an argument could be made for a weak wavetrain, but unlike Figure 6 there is no clear separation between a leading wave form and the trailing waves. In the highest velocity case the remnants of the rightward propagating wavetrain trail the disturbances near the surface by at least 1000 m.

Since background shear has been found to possibly affect the polarity of ISWs, we performed numerical experiments with an initial condition with reversed polarity (a depression). The results of polarity reversal are compared for $U_0 = 0.3$ m s$^{-1}$ in

Figure 12. Figure 12a reproduces Figure 11b, while Figure 12b shows the polarity reversed case. It can be seen that the leftward propagating wave in panel (b) takes the form of an undular bore, consistent with the predictions of WNL theory. The undular bore is much slower than all ISWs simulated. Interestingly, the rightward propagating wave train also yields an undular bore,

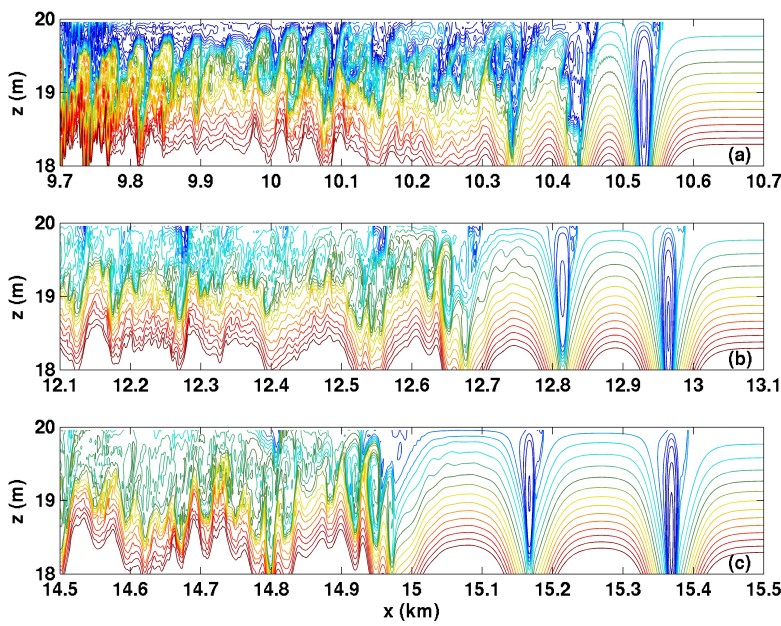

**Figure 7.** Detail of density fields in the core region for wavetrains propagating with the shear current for the case NO DJL 1. 21 density isolines with values $-5 \times 10^{-4} < \rho < -4 \times 10^{-4}$ shown. (a) $t = 7,200$ s, (b) $t = 14,400$ s, and (c) $t = 21,600$ s.

however for later times (not shown) the undular bore largely fades away to irrelevance, and the disturbances of the high shear region dominate the rightward propagating response.

The cases described in this subsection, are perhaps the clearest demonstration of the difference between the weak shear regime discussed in Stastna and Lamb (2002), for which ISWs are modulated in form by the background shear current, and the strong background shear regime.

## 3   Discussion

The numerical experiments reported above illustrate that when a strong background shear current is present the set of possible
wave-vortex phenomena is considerably larger than the tidy ISWs described by the DJL equation. Indeed essentially none of the phenomena we have simulated could be termed as truly steady. The main coherent feature from our simulations one could expect to see in the field for strong background shear current are ISWs with a strong vortex core in the near surface, high shear region. Classical theory of ISWs with cores generally considers nearly quiescent cores (Derzho and Grimshaw (1997); Luzzatto-Fegiz and Helfrich (2014); Lamb and Wilkie (2004)), while recent theory and simulations that includes background
shear currents yields so–called 'subsurface cores' (He et al. (2019)). Neither of these matches the strong wave-vortex coupling

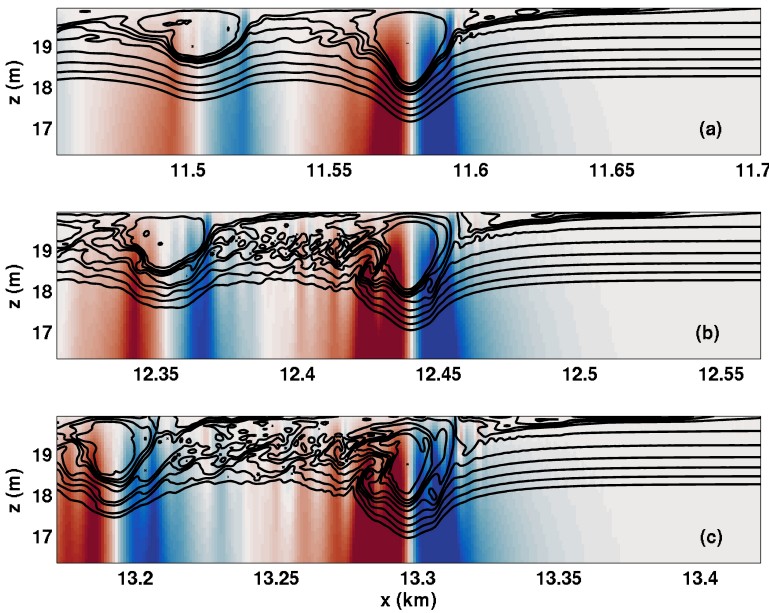

**Figure 8.** Detail of vertical velocity (shaded and saturated at $\pm 0.01$ m s $^{-1}$) and density fields in the core region for wavetrains propagating with the shear current critical layer case CL 1. 10 density isolines with $-5 \times 10^{-4} < \rho < -4 \times 10^{-4}$ shown in black. (a) $t = 7,200$ s, (b) $t = 9,000$ s, and (c) $t = 10,800$ s

that was observed above (e.g., Figure 7). In cases in which a critical layer is possible, the vortex cores coexist with smaller disturbances that extend ahead of the leading ISW.

The simulations we performed required resolution that was much higher than that typically used in coastal ocean simulations (i.e., multi-kilometer domains with horizontal resolution of less than a meter and vertical resolution less than 5 cm). Even

recent process studies on mode-2 waves in the presence of shear using the finite volume MITgcm (Zhang et al. (2018)) used a resolution of $5 \times 0.5$ m (which was more than sufficient for the phenomena these authors set out to model). Moreover, the pseudospectral model used has very low inherent numerical dissipation so that even phenomena with relatively few grid points across them are only weakly damped. The high resolution necessary makes extension to three dimensions a complicated task. However, 3D simulations initialized from a two dimensional simulation, i.e. using the early portion of the simulations described

above, should be possible with present computational resources. It is likely that performing a detailed energy analysis, as in Lamb (2010) for ISWs propagating against the background shear current (hence ones with a DJL theory), would require 3D simulations.

However, a more serious issue when interpreting the above results is in the large gap between theory inspired simulations and those on the regional scale in coastal oceans. To the authors' knowledge there has been no systematic study of ISW behaviour

in marginally resolved situations, especially when parameterizations for mixed layers (e.g., the KPP scheme) are employed. As


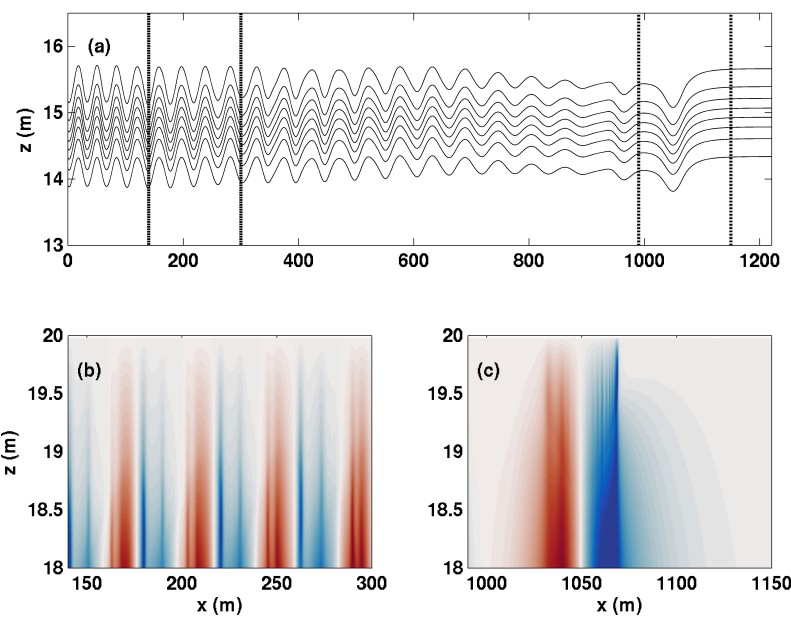

**Figure 9.** (a) Detail of density fields for wavetrains propagating with the shear current critical layer case CL 2 (the case with a narrow density profile) $t = 4500$ s, (b) the vertical velocity component in the top 2 m of the domain in the region between the thick black lines in panel (a) saturated at $\pm 0.005$ m s$^{-1}$, (b) the vertical velocity component in the top 2 m of the domain in the region between the thin dotted black lines in panel (a), saturated at $\pm 0.005$ m s$^{-1}$.

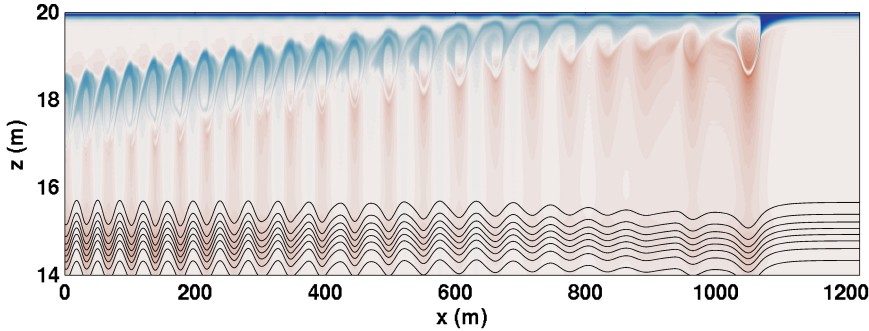

**Figure 10.** The relative vorticity field staurated at $0.1$ s$^{-1}$ for the CL2 case with the density field overlaid at $t = 4500$ s.

noted above, the key feature of ISWs is their ability to outrun local instabilities or mixing events without a loss of coherence. A globally high value of eddy viscosity cannot be 'outrun' and thus care should be taken in extrapolating simulation results to particular field measurements.


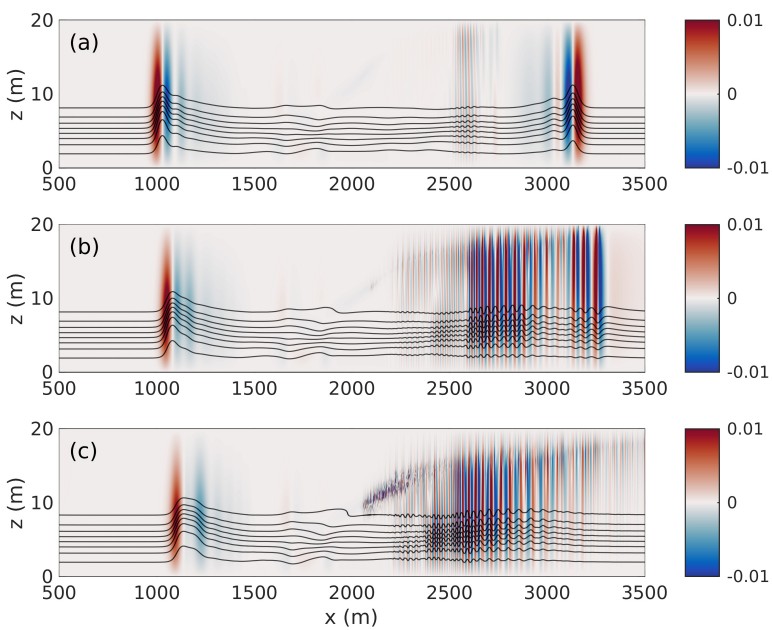

**Figure 11.** The vertical component of velocity (shaded) saturated at $\pm 0.01$ m s$^{-1}$ with 8 isolines of density superimposed at $t = 5400$ s. (a) case DJLB, (b) case NO DJL 1B, (c) case CL 1B.

Even with the above caveats, the simulations we have conducted pose a very clear question for the theorist: "In deep water
in which a pycnocline may be far removed from local shear layers can an ISW be destabilized by a shear that would be thought
of as so far from the wave that it would be irrelevant to its evolution?"

## 4   Conclusions

We have provided one possible explanation for why the measurements in Walter et al. (2016) yielded large amplitude waves
with no DJL based description. Namely, that for a portion of parameter space, mode-1 ISWs generated via stratified adjustment
in the presence of a background shear current are not free waves, since they have a propagation speed very close to the
maximum value of the background current. When some stratification is observed near the surface, these waves take the form
of ISWs with strongly recirculating cores.

The simulations above were two dimensional, while the observations of Walter et al. (2016) were influenced by rotation, and
likely include some large scale three-dimensional structures (comments on small scale three-dimensionalization were made in
the Discussion, above). Moreover, the observations take a different form from the wave trains that are spontaneously generated
in our stratified adjustment simulations. This is not surprising, and a set of initial conditions more closely tied to the field
observations remain a clear goal for future work.


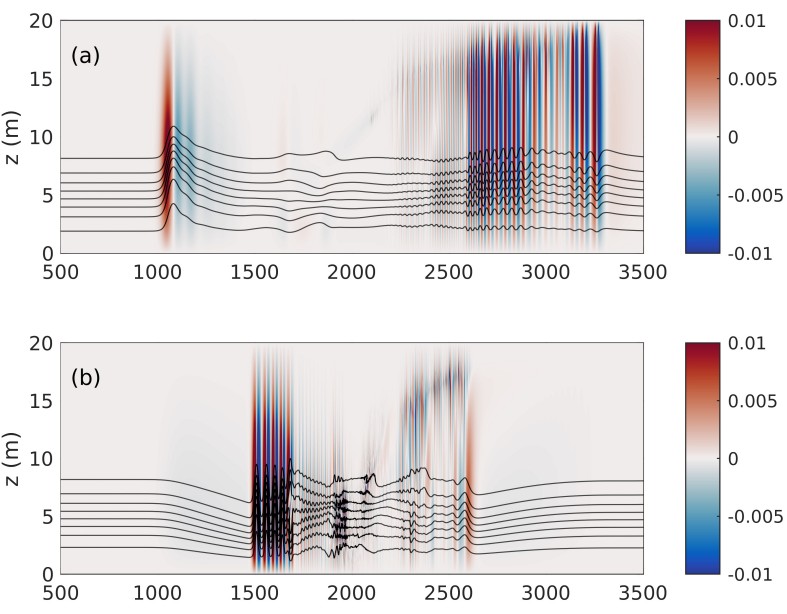

**Figure 12.** The vertical component of velocity (shaded) saturated at $\pm 0.01$ m s$^{-1}$ with 8 isolines of density superimposed at $t = 5400$ s. (a) case NO DJL 1B, (b) case NO DJL 1BF (polarity reversal).

A different avenue for future work would be to more closely link our results to the work of Zhang et al. (2018) on mode-2 waves. Mode-2 waves lack a DJL theory due to the possible presence of a trailing tail of mode-1 waves, and the question of 305 whether a particular shear current more strongly affects the leading mode-2 wave or its mode-1 tail remains largely open.

Finally we provided well resolved examples of stratified adjustment with critical layers (cases labelled 'CL'). When a significant stratification was present near the surface, the waves formed consisted of strong vortex cores, with some evidence of fine scale structure on the leading side of the wave. In the absence of stratification near the surface, a train of vortices spontaneously formed near the surface and this led to small amplitude deformations of the underlying pycnocline. Our simulations did not 310 yield evidence for the common theoretical assumption that ISWs are the dominant object, with critical layers providing a slow drain of energy (Caillol and Grimshaw (2012)). This observation should be re-examined on scales for which three-dimensional DNS is possible.

## 5 Code and data availability

The code used, SPINS, is publicly available through github and its webpage:

https://wiki.math.uwaterloo.ca/fluidswiki/index.php?title=SPINS



The case files for all numerical experiments reported on above are available from the corresponding author, as are the relevant data sets.

*Author contributions.* AC carried out exploratory simulations, contributed to the design of figures and participated in the writing of the manuscript. RW provided access to relevant oceanographic data sets, commented on both exploratory and final simulations and contributed

large parts of the manuscript organization. MS designed all final simulations and did the majority of the writing and figure construction.

*Competing interests.* NONE

*Acknowledgements.* This research was supported by the Natural Science and Engineering Research Council of Canada through a Doctoral scholarship for AC, and Discovery Grant RGPIN-311844-37157 for MS.





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
