# Peer review of "The effect of strong shear on internal solitary-like waves"

_Nonlinear Processes in Geophysics, 2021_

## Author Response (AR1)

We thank both reviewers for their helpful and constructive comments. Below the reviewer comments are given with our reply.

Reviewer 1:

The refereed paper is devoted to the description of the series of numerical experiments performed with high resolution on simulations of wavefield adjustment in a stratified basin with strong shear flows. It is demonstrated that strongly nonlinear large-amplitude solitary-like waves of bell-shaped forms can stably propagate in the counter-current direction. In the meantime, co-current propagating perturbations have recirculating vortex cores completely different shapes from the classical theory of internal solitary waves. Moreover, behind the leading waves, a turbulent wake is generated and gradually separates from the frontal waves. Both co-current and counter-current waves are spontaneously generated from general initial conditions demonstrating further very asymmetric behaviour. The cases associated with critical layers were also observed. The authors discuss the application of the numerical experiments to the real observations in situ and to existing theories; the results obtained provide a partial reconciliation between observations and theory.

The paper is clearly written in good English, contains new, important, and interesting results. I am recommending the paper for publication in the journal with only minor remarks and suggestions.

1) I would suggest citing papers in the same references in chronological order. In particular, on page 1, line 15, it would be logical to refer to the papers in the reverse order (Talipova et al. (1999); Helfrich and Melville (2006)) rather than (Helfrich and Melville (2006); Talipova et al. (1999)).

Reply: We have adopted the reviewer's suggestion in this instance, as well as others in the manuscript.

2) On the same page 1, line 15 it would be reasonable to add the

reference on the review: Ostrovsky et al., Chaos, 2015, DOI: 10.1063/1.4927448

 Reply: Thank you for the interesting reference, which we have added to the manuscript.

3) On page 3, line 79 extra 'the' must be removed.

 Reply: we have done this

4) On page 4, line 100 I am suggesting replacing the term 'shear' viscosity with the term 'molecular kinematic' viscosity.

 Reply: we have done this

5) I was going to remark that the use of the molecular kinematic viscosity is doubtful in the context of water waves in such a rather big basin, but the authors have anticipated my remark and mentioned the importance of the eddy viscosity on lines 287–288. This is indeed an important issue that can be studied separately.

 Reply: we thank the reviewer for their supportive comment.

6) It would be good to clarify in a bit more details the physics of such a big difference and asymmetry in the ISW behaviour when they propagate co-current and counter-current. It is amazing that despite the wave amplitudes in the latter case are notably higher, they stably propagate, whereas in the former case, the instability occurs producing a turbulent wake.

Reply: we have expanded the discussion related to this comment at several points in the manuscript.  The two main new points are: i) a link to the KdV theory through changes in the nonlinear and dispersion coefficients, ii) a breakdown of the instability types that occur into categories, with a modified figure of the Richardson number that shows how this quantity evolves.

Reviewer 2:

In this paper numerical experiments are used to investigate the effect of background shear on internal solitary waves (ISWs). Cases of wave propagation both with and against a background shear current (travelling from left to right) are considered by using a numerical domain in which a stratified adjustment problem in the centre of the domain generates both leftward (against shear) and rightward (with shear) wave propagation. The waves are classified with respect to classical DJL theory. The paper shows that ISWs generated by stratified adjustment in the presence of background shear are not necessarily free waves and can differ markedly from DJL solutions. In particular they can have a propagation speed close to the maximum value of the background current and can exhibit strong recirculating cores.

The paper raises and asks important questions on how classical theory and simulations may be linked to the field; it highlights the need for more investigation in this area. The work is original, clearly presented and of significant interest to communities including physical oceanographers and fluid modellers. As such I have no hesitation in highly recommending this paper for publication in NPG subject to the minor comments/revisions given below.

line 60 typo and and

Reply: we have fixed the typo

A figure/graphic showing the initialisation of the numerical simulations would aid the reader. I didn't fully understand the stratification adjustment until I saw fig 3(a).

Reply: a new figure of the initialisation has been added

fig 2 and fig 3 - these are really interesting results. Can the authors give a physical explanation why the leftward propagating waves (those going against the shear) are steeper, shorter and have larger amplitude than the right ward ?

Reply: this was identified as a point needing clarification by both reviewers and hence has had a discussion added at several point. Links to the KdV theory have been made, via the dependence of the

nonlinearity and dispersion coefficients when the background current changes.

fig 4 (a) The N^2 plot is interesting but doesn't show anything that can't be discerned from 4(b) and as such I wonder if illustrating density (like fig 3) would be more appropriate ?

Reply: after some thought we agree with the reviewer, and have chosen to show the Ri field for two different times. This allowed us to expand the discussion of the types of instability observed in a new paragraph. Much appreciated!

fig 6. The authors say these waves look more like Stokes waves than typical solitary waves - can they give more justification or an appropriate reference. Could a comparison be drawn to cnoidal waves ?

Reply: we have added a reference for cnoidal waves (there are others, but this one seemed the clearest). It is somewhat subjective whether a reader sees the resemblance to cnoidal waves or Stokes waves (turned upside down). The latter is perhaps more generally known, so we have kept the term, adding cnoidal waves as well and focusing the reference on the latter, more thoroughly documented case.

fig 7. This is fascinating - what causes the instability in the top layer ? The waves are out running the manifestation of the instability so if the waves are not the cause (or continual cause) what is ? Is the rigid lid in the simulations important ?

Reply: This has been commented on using the new version of the Ri figure (see response above). There are three instability types, two of which are visible in this figure. One is a stratified shear instability triggered by the finite amplitude perturbations due to the developing wave train (i.e. the ISW-like waves take time to sort from the perturbations, so there is enough energy to trigger a very busy field of instabilities), the other is the vortex cores of the emerging ISW-like waves.

fig 9 caption ( c) is not detailed

Reply: this has been fixed in the revised manuscript

233 typo form - from

Reply: the typo has been fixed